# Comparison across age groups of causes, circumstances, and consequences of falls among individuals living in Canada: A cross-sectional analysis of participants aged 45 to 85 years from the Canadian Longitudinal Study on Aging

**Vanina P. M. Dal Bello-Haas**[1]*, **Megan E. O'Connell**[2], **Jake Ursenbach**[2]

1 School of Rehabilitation Science, McMaster University, Hamilton, Ontario, Canada, 2 Department of Psychology & Health Studies, University of Saskatchewan, Saskatoon, Saskatchewan, Canada

☯ These authors contributed equally to this work.

* vdalbel@mcmaster.ca

## Abstract

Falls are a leading cause of injury-related deaths and hospitalizations among Canadians. Falls risk has been reported to be increased in individuals who are older and with certain health conditions. It is unclear whether rurality is a risk factor for falls. This study aimed to investigate: 1) fall profiles by age group e.g., 45 to 54 years, 55 to 64 years, 65 to 74 years, 75 to 85 years; and 2) falls profiles of individuals, by age group, living in rural versus urban areas of Canada. Data (N = 51,338) from the Canadian Longitudinal Study on Aging was used to examine the relationship between falls and age, rurality, chronic conditions, need for medical attention, and fall characteristics (manner, location, injury). Self-reported falls within a twelve-month period occurred in only 4.8% (single fall) and 0.8% (multiple falls) of adults. Falls were not related to rural residence or age, but those with memory impairment, multiple sclerosis, as well as other chronic conditions such as mood disorder, anxiety disorder, and hyperthyroidism not often thought to be associated with falls, were also more likely to fall. Older individuals were more likely to fall indoors or fall while standing or walking. In contrast, middle-aged individuals were more likely to fall outdoors or while exercising. Type of injury was not associated with age, but older individuals were more likely to report hospitalization after a fall. This study shows that falls occur with a similar frequency in individuals regardless of age or urban/rural residence. Age was associated with fall location and activity. A more universally applicable multi-facted approach, rather than one solely based on older age considerations, to screening, primary prevention and management may reduce the personal, social, and economic burden of falls and fall-related injuries.

**Data Availability Statement:** Data are available from the Canadian Longitudinal Study on Aging (www.clsa-elcv.ca) for researchers who meet the criteria for access to de-identified CLSA data.

**Funding:** McMaster Institute of Research on Aging (MIRA) for Canadian Longitudinal Study on Aging (CLSA) data access. Funding for the Canadian Longitudinal Study on Aging (CLSA) is provided by the Government of Canada through the Canadian Institutes of Health Research (CIHR) under grant reference: LSA 94473 and the Canada Foundation for Innovation, as well as the following provinces, Newfoundland, Nova Scotia, Quebec, Ontario, Manitoba, Alberta, and British Columbia. The funders had no role in study design, data collection and analysis, decision to publish, or preparation of the manuscript.

**Competing interests:** Co-author Megan O'Connell is a Member - Psychology Working Group, Canadian Longitudinal Study on Aging (CLSA) https://www.clsa-elcv.ca/.

## Introduction

Falls and understanding fall risk are important national health care priorities and public health concerns for Canadians. One third of Canadians fall each year [1], with fall rates increasing with increasing age [2, 3]. Prevention of falls is critical to keeping people independent and active across the lifespan, and particularly in later life. Falls have been found to be associated with an array of risk factors [4–6], and result in a plethora of serious consequences including loss of independence, disability, decline in health and well-being, decreased quality of life, pain, morbidity, institutionalization, and mortality [2, 3, 7]. Almost half of Canadians ages 20 years and older are living with a chronic condition [8], which can put them at increased risk of falls [9]. Falls are the leading cause of injury-related deaths and hospitalizations among all Canadians. Direct and indirect costs of falls are staggering, and societal and personal burden are significant [7, 10, 11].

Canada's population continues to age [12]. Canada's rural population is aging faster than its urban and suburban counterparts, with Canadians aged 65 and over comprising 18% of Canada's rural population [13]. Social and built environment characteristics shape opportunities for and barriers to health promotion [14, 15], and evidence is compelling that social and built environmental characteristics affect health [16, 17]. Research on the connections between community-level factors and health has shown burden of illness to be greater for vulnerable populations, such as older adults [18] and people living in rural and remote areas [18–20].

With the expected accelerating pace of aging in Canada, health declines and challenges, such as falls and their related consequences, carry profound implications and highlight the need to understand when falls present and how they change across the lifespan. Understanding falls and fall risk can inform the timing of programs, services, and public health interventions. The reported high incidence of falls that occur with aging, as well as fall-related mortality and morbidity, underscores the importance of preventive measures and interventions for Canadian adults. No research to date has systematically and comprehensively examined age-group differences in mobility, mobility limitation, and fall profiles in Canadians, nor has any research to date examined differences in mobility, mobility limitations, and fall profiles of adults living in rural versus urban Canada. The objectives of our research were to: 1) comprehensively examine fall profiles by age group e.g., 45 to 54 years, 55 to 64 years, 65 to 74 years, 75 to 85 years; 2) compare falls profiles of adults, by age group, living in rural versus urban areas of Canada; and 3) identify any key issues specific to the rural setting (social, built environment) that affect number of falls.

## Materials and methods

This was a cross-sectional study, analyzing population-based data from the Canadian Longitudinal Study on Aging (CLSA). A protocol for the planned analysis was submitted the CLSA Data and Sample Access Committee and the University of Saskatchewan Ethics Committee (Beh #250) for approval prior to accessing the data and analysis. Data were accessed on November 13. 2021 for research purposes. At no time did the authors have access to information that could identify individual participants during or after data collection.

### Study design and participants

Details on the CLSA study design and methods have been described elsewhere and are briefly outlined here [21, 22]. The CLSA is a large, national, long-term study that will follow for at least 20 years more than 50,000 men and women ages 45 to 85 years at enrolment. Individuals were not eligible to participate if: they were living in one of Canada's three territories, on First Nation reserves, or in long-term care facilities; were full-time members of the armed forces; were unable to communicate in English or French; or, had overt cognitive deficits. At CLSA baseline 21,241 participants were randomly selected from the 10 Canadian provinces and completed

telephone-based questionnaires (i.e., Tracking cohort). Another 30,097 participants were randomly selected from areas extending 25 to 50 kilometres from each of 11 Data Collection Sites located across Canada and provided data by visiting a Data Collection Site and through an in-home interview (i.e., Comprehensive cohort). The aim of the CLSA was to explore multifaceted aspects of aging, and questionnaire/interview/assessment data included information about socioeconomic status, demographics, participation in daily activities, mental health, and objective and subjective measures of cognitive and physical health. Tracking cohort data were collected between 2011-09-22 and 2014-05-03, and the Comprehensive data between 2011-12-05 to 2015-07-07. The Tracking database has a larger proportion of rural Canadians, and the Comprehensive database includes all questionnaires asked of the Tracking cohort but includes additional measures, including physical measurement variables. For the current project, cross sectional data from the baseline Tracking and Comprehensive cohorts were combined, and analyses were restricted to variables measured equivalently across each cohort.

## Measures

**Demographic data.** Binary sex (Male, M; Female, F). education level, marital status, and yearly income were described for the sample. Age was collected and reported as a continuous variable, but for analyses we grouped participants into the following age groups: 45–54, 55–64, 65–74, 75+ years. Categorization of residence for each participant was based on the forward sorting area from postal codes and categorized as rural, secondary urban core, urban fringe centre, and urban, per Statistics Canada classifications [23]. Rural refers rareas of census metropolitan areas (CMAs) and census agglomerations (CAs), formed by one or more adjacent municipalities centered on a population center (urban core), as well as population living in rural areas outside CMAs and CAs. A CMA total population is at least 100,000 of which 50,000 or more live in the core. A CA core population is at least 10,000. A CMA or CA can have: 1) the core, population centre with the highest population, around which a CMA or a CA is delineated; and 2) the secondary core, a population centre within a CMA that has at least 10,000 persons. Urban fringe includes apopulation centres within a CMA or CA that have fewer than 10,000 persons and are not contiguous with the core or secondary core.

**Medical conditions.** Participants were asked whether a doctor has told them they have a chronic condition (i.e., a transient ischemic attack, stroke, memory problems, dementia or Alzheimer's disease, Parkinson's disease, multiple sclerosis, epilepsy, cancer).

**Fall data.** In the maintaining contact questionnaire, which was asked approximately 18 months after the baseline interview, participants were asked "how many times have you fallen in the past 12 months?" For the current study, participants were grouped as follows: those who reported no fall history (no falls), those who reported a single fall, (single fall) and those who reported two or more falls (multiple falls). Additional fall-related data were collected only for participants who reported one or more falls (i.e., fallers). Fallers were asked to focus on their most serious injury from a fall and were given the following options to describe the nature of their injury: no serious injury, sprain/strain, bruises, cuts, discomfort, hip fracture, leg fracture, arm/wrist fracture, vertebral fracture, head injury, or other. Fallers were asked whether they sought medical attention from a health professional within 48 hours following a fall (yes/no) and whether they were hospitalized due to this fall (yes/no). Participants were then given a list of options to describe where the fall happened i.e., inside home, outside home but inside a building, or outdoors. Finally, participants were provided with the following list to describe how the fall happened: fell while standing/walking, fell on steps/stairs, fell while exercising, fell from height of 1 metre, fell from furniture, fell while getting in/out of bath, fell while getting in/out of shower, fell on snow or ice, or other.

## Statistical analyses

Analyses were completed using the Statistical Package for Social Sciences (IBM SPSS®, version 28). Descriptive statistics included mean, standard deviation (SD) and frequency counts; and group-based differences based on age group, sex, or rural/urban status were compared with ANOVAs or chi-square analyses where appropriate.

Binary logistic regression was used to explore whether demographic factors (age measured as a continuous variable, sex, education level, rural/urban) were predictors of fall status, that is no falls versus a single fall or multiple falls (reference group 'no falls'). We used sampling weights (version 1.2) [24] that were adjusted for the Canadian population to explore if this impacted the findings. Due to similar findings between weighted and unweighted results, we only reported the unweighted results. Sampling weights inflate the observations in the sample to the level of the population to minimize the sampling bias, allowing observations within the sample to be extrapolated to the population of origin. The large sample size for many of these analyses and large number of comparisons were likely to inflate Type I error [25]. For associations we focussed on the magnitude of effect. The following magnitude descriptors were used for associations between dichotomous variables: phi = 0.00 to <0.01, negligible; 0.10 to < 0.20, small; 0.20 to <0.40, moderate; 0.40 to <0.60, relatively large; 0.60 to <0.80, large; 0.80 to 1.00, very large [26]. For the odds ratio (OR) estimates from the logistic regressions, we used the descriptors of magnitude of OR provided by Chen et al. [27] based on a rate of falls of around 5% of OR ~ 1.5 as small, OR ~ 2.7 as medium, and OR ~ 4.6 as large.

## Results

Participant demographics are described in Table 1. Of the 46,762 participants who completed the initial falls related question, "how many times have you fallen in the past 12 months?", the vast majority reported no falls (94.4%) and a further 4.8% reported a single fall in during the preceding 12 months. Very few participants reported multiple falls (0.8%), with the number of falls ranging from 2 to 30 falls. Table 2 includes information about falls (no fall, single fall, multiple falls) with age group comparisons. Age group was not significantly associated with fall status in the smaller sized rural dwelling sample ($N$ = 6,198) but was associated with fall status for urban dwelling participants, likely a Type I error due to the large $N$ (35,779) for the urban sample. Younger participants in the urban-dwelling group were less likely to be in the multiple faller group and older participants were more likely to be the multiple faller group, but these differences were rather small (phi < 0.1). As there were no substantial differences between the urban and rural dwelling groups, no further results by location are reported.

Age group was not associated with whether participants sought medical attention after a fall ($p$ = 0.38). While the chi-square was significant ($p$ = 0.011) for the single fall group, the effect size *size* (phi = 0.071) was trivial in magnitude (Table 3). In contrast age group was weakly associated with hospitalization after a fall, with participants in the older age groups more likely to report being hospitalized than younger participants. The effect size estimate was small in magnitude for both the single fall (phi = 0.12) and multiple falls groups (phi = 0.14), but only the single fall group had a significant chi-square ($p$ < 0.001). A small magnitude association (single falls phi = 0.11; multiple falls phi = 0.18) between where falls occurred and age group was evident (Table 4), with older participants more likely to fall inside the home than outside the home relative to younger participants.

Age group was associated (moderate effect size) with how falls occurred for both the single fall group (phi = 0.24) and the multiple fall group (phi = 0.30) (Table 5). The most pronounced age group differences were for falls while exercising. Not surprising, older participants were less likely to fall during exercise and younger participants were more likely to fall during

**Table 1. Sample characteristics (N = 51,338).**

| Demographic Variable | M (SD) or Count'(%) | Missing |
|---|---|---|
| Age | 62.98 (10.43) | |
| Sex | | |
| • F | 26155 (50.9%) | |
| • M | 25183 (49.1%) | |
| Education Level | | 133 (0.3%) |
| • Less than secondary school graduation | 3629 (7.1%) | |
| • Secondary school graduation, no post-secondary education | 5721 (11.1%) | |
| • Some post-secondary education | 3861 (7.5%) | |
| • Post-secondary degree/diploma | 37994 (74.0%) | |
| Residence—Rural/Urban | | 2503 (4.9%) |
| • Rural | 7131 (13.9%) | |
| • Urban Core, Fringe, Centre, Secondary Core | 41704 (81.2%) | |
| Marital Status | | 14 (0.0%) |
| • Single/Never Cohabitated | 4352 (8.5%) | |
| • Married/Common-law | 35252 (68.7%) | |
| • Widowed | 5170 (10.1%) | |
| • Divorced | 5180 (10.1%) | |
| • Separated | 1370 (2.7%) | |
| Total Yearly Income (Canadian Dollars) | | 3311 (6.5%) |
| • Less than $20,000 | 2913 (5.7%) | |
| • $20,000 to $50,000 | 12209 (23.8%) | |
| • $50,000 to $100,000 | 17127 (33.4%) | |
| • $100,000 to $150,000 | 8739 (17.0%) | |
| • $150,000 or more | 7039 (13.7%) | |
| Diagnosed with a Neurological Condition | | |
| • No | 27786 (54.1%) | |
| • Yes | 23552 (45.9%) | |
| Multiple Falls | | 4576 (8.9%) |
| • No | 46406 (90.4%) | |
| • Yes | 356 (0.7%) | |
| Number of Falls | | 4576 (8.9%) |
| • No falls | 44164 (86.0%) | |
| • Single fall | 2242 (4.4%) | |
| • Multiple falls | 356 (0.7%) | |
| High blood pressure or hypertension | 19203 (37.4%) | 207 (0.4%) |
| Diabetes, borderline diabetes or blood sugar is high | 8863 (17.3%) | 129 (0.3%) |
| Underactive thyroid gland (hypothyroidism) | 6408 (12.5%) | 569 (1.1%) |
| Overactive thyroid gland (hyperthyroidism) | 1190 (2.3%) | 568 (1.1%) |
| Asthma | 6331 (12.3%) | 166 (0.3%) |
| Osteoarthritis in one or both hands | 6840 (13.3%) | 386 (0.8%) |
| Osteoarthritis in the hip | 4586 (8.9%) | 416 (0.8%) |
| Osteoarthritis in the knee | 7927 (15.4%) | 458 (0.9%) |
| Rheumatoid arthritis | 2058 (4.0%) | 516 (1.0%) |
| Other type of arthritis | 6292 (12.3%) | 503 (1.0%) |
| Osteoporosis | 4698 (9.2%) | 341 (0.7%) |
| Cancer | 7902 (15.4%) | 109 (0.2%) |
| Intestinal or stomach ulcers | 3912 (7.6%) | 171 (0.3%) |

(*Continued*)

**Table 1.** (Continued)

| Demographic Variable | M (SD) or Count'(%) | Missing |
|---|---|---|
| Urinary incontinence | 4388 (8.5%) | 125 (0.2%) |
| Bowel incontinence | 1074 (2.1%) | 105 (0.2%) |
| Bowel disorder | 4776 (9.3%) | 171 (0.3%) |
| Kidney disease or kidney failure | 1460 (2.8%) | 160 (0.3%) |
| Lung-related disorder | 3161 (6.2%) | 202 (0.4%) |
| Heart disease | 5694 (11.1%) | 228 (0.4%) |
| Heart attack or myocardial infarction | 2778 (5.4%) | 179 (0.3%) |
| Angina | 2473 (4.8%) | 190 (0.4%) |
| Peripheral vascular disease or poor circulation in limbs | 3165 (6.2%) | 230 (0.4%) |
| Macular degeneration | 2155 (4.2%) | 257 (0.5%) |
| Experienced a ministroke or TIA | 1713 (3.3%) | 306 (0.6%) |
| Migraine headaches | 6773 (13.2%) | 136 (0.3%) |
| Memory problem | 968 (1.9%) | 113 (0.2%) |
| Mood disorder | 8250 (16.1%) | 129 (0.3%) |
| Anxiety disorder | 4157 (8.1%) | 146 (0.3%) |
| Other long-term physical or mental condition | 21103 (41.1%) | 118 (0.2%) |

exercise. Older participants were more likely to report falling while standing or walking than younger participants in both the single and multiple falls group. Older participants who reported a single fall were more likely to report falling on snow or ice than were younger participants, but the converse was apparent for participants in the multiple falls group, with younger participants more likely to report falling on snow or ice. Older participants who were in the single fall group were more likely to report falling from furniture, however reported falls getting in and out of the shower and bath were infrequent regardless of age group Table 6 includes the type of injury experienced during the most serious fall experienced by participants, and these were not significantly associated with age group.

Odds ratios for single and multiple falls for participants with chronic conditions are reported in Table 7. The likelihood of having multiple falls was substantially higher (i.e., large magnitude of effect) for participants who reported being diagnosed with memory problems or with multiple sclerosis. Moderately higher likelihood of having multiple falls was also apparent for participants diagnosed with osteoarthritis in the hands, asthma, bowel disorder, anxiety

**Table 2. Number of falls by age group and rural/urban residence (N = 44,455).**

| Dwelling | AgeGroup(years) | Number of Falls | | | |
|---|---|---|---|---|---|
| | | No Falls | Single Fall | Multiple Falls | Total |
| Rural | 45–54 | 1725 (95.3%) | 74 (4.1%) | 11 (0.6%) | 1810 (100.0%) |
| | 55–64 | 2010 (94.9%) | 95 (4.5%) | 14 (0.7%) | 2119 (100.0%) |
| | 65–74 | 1484 (94.9%) | 74 (4.7%) | 5 (0.3%) | 1563 (100.0%) |
| | 75+ | 979 (94.0%) | 52 (5.0%) | 11 (1.1%) | 1042 (100.0%) |
| | **Total** | **6198 (94.9%)** | **295 (4.5%)** | **41 (0.6%)** | **6534 (100.0)** |
| Urban (Urban Core, Fringe, Centre, Secondary Core) | 45–54 | 8755 (94.7%) | 429 (4.6%) | 59 (0.6%) | 9243 (100.0%) |
| | 55–64 | 11267 (94.0%) | 609 (5.1%) | 114 (1.0%) | 11990 (100.0%) |
| | 65–74 | 8590 (94.7%) | 420 (4.6%) | 62 (0.7%) | 9072 (100.0%) |
| | 75+ | 7167 (94.1%) | 386 (5.1%) | 63 (0.8%) | 7616 (100.0%) |
| | **Total** | **35779 (94.4%)** | **1844 (4.9%)** | **298 (0.8%)** | **37921 (100.0%)** |

**Table 3. Frequency of falls requiring medical attention by age group and single/multiple falls (N = 2,566).**

| Number of Falls | AgeGroup (years) | Fall Required Medical Attention | | |
|---|---|---|---|---|
| | | **Yes** | **No** | **Total** |
| Single Fall | 45–54 | 321 (60.9%) | 206 (39.1%) | 527 (100.0%) |
| | 55–64 | 442 (60.8%) | 285 (39.2%) | 727 (100.0%) |
| | 65–74 | 324 (63.5%) | 186 (36.5%) | 510 (100.0%) |
| | 75+ | 313 (69.7%) | 136 (30.3%) | 449 (100.0%) |
| | **Total** | **1400 (63.3%)** | **813 (36.7%)** | **2213 (100.0%)** |
| Multiple Falls | 45–54 | 36 (50.7%) | 35 (49.3%) | 71 (100.0%) |
| | 55–64 | 80 (59.7%) | 54 (40.3%) | 134 (100.0%) |
| | 65–74 | 45 (64.3%) | 25 (35.7%) | 70 (100.0%) |
| | 75+ | 48 (61.5%) | 30 (38.5%) | 78 (100.0%) |
| | **Total** | **209 (59.2%)** | **144 (40.8%)** | **353 (100.0%)** |

Multiple Falls Group: two or more falls reported in preceding year (No or Yes)

disorder, angina, rheumatoid arthritis, transient ischemic attack, kidney disease, bowel incontinence, and for the few participants who reported diagnoses of Parkinson's disease or Alzheimer's disease. For the following other chronic conditions there was an increased likelihood of reporting a single fall or multiple falls, with the magnitude of effect stronger for the multiple fall vs single fall groups: mood disorders, osteoarthritis of the knee, osteoarthritis of the hip, osteoporosis, urinary incontinence, lung related disorders, stroke, and epilepsy. Other significant associations (i.e., where the OR CI did not include 1) were small or trivial.

## Discussion

Our study has contributed important information regarding the incidence of falls and fall-related injuries in community-dwelling Canadians ages 45 to 75+ years. We sought to explore rural–urban location as a factor to advance the understanding of falls and fall rates in high-income countries. We hypothesized there may be urban-rural differences, as disparities in health status [11, 28, 29], risk factors associated with poorer health including physical inactivity [30–32], and barriers to accessing and utilizing health services and facilities and health

**Table 4. Frequency of falls in different locations by age group and single/multiple falls (N = 2,592).**

| Number of Fallls | Age Group (years) | Fall Location | | | |
|---|---|---|---|---|---|
| | | **Inside home** | **Outside Home but Inside a Building** | **Outdoors** | **Total** |
| Single Fall | 45–54 | 92 (17.4%) | 101 (19.1%) | 337 (63.6%) | 530 (100.0%) |
| | 55–64 | 133 (18.1%) | 140 (19.1%) | 461 (62.8%) | 734 (100.0%) |
| | 65–74 | 121 (23.4%) | 100 (19.4%) | 295 (57.2%) | 516 (100.0%) |
| | 75+ | 133 (29.2%) | 85 (18.6%) | 238 (52.2%) | 456 (100.0%) |
| | **Total** | **479 (21.4%)** | **426 (19.1%)** | **1331 (59.5%)** | **2236 (100.0%)** |
| Multiple Falls | 45–54 | 13 (18.1%) | 11 (15.3%) | 48 (66.7%) | 72 (100.0%) |
| | 55–64 | 41 (30.4%) | 16 (11.9%) | 78 (57.8%) | 135 (100.0%) |
| | 65–74 | 28 (40.0%) | 5 (7.1%) | 37 (52.9%) | 70 (100.0%) |
| | 75+ | 30 (38.0%) | 10 (12.7%) | 39 (49.4%) | 79 (100.0%) |
| | **Total** | **112 (31.5%)** | **42 (11.8%)** | **202 (56.7%)** | **356 (100.0%)** |

Multiple Falls Group: two or more falls reported in preceding year (No or Yes)

**Table 5. Frequency of falls in different manners by age group and single/multiple falls (N = 2551).**

| Number of Falls | Age Group (years) | Fall Manner | | | | | | | | | |
|---|---|---|---|---|---|---|---|---|---|---|---|
| | | Fall while standing or walking | Fell on stairs or steps | Fell while exercising | Fell from height of 1 metre | Fell from furniture | Fell while getting on or out of bath | Fell while getting in and out of the shower | Fell on snow or ice | Other | Total |
| Single Fall | 45–54 | 198 (37.8%) | 103 (19.7%) | 117 (22.3%) | 39 (7.4%) | 6 (1.1%) | 2 (0.4%) | 3 (0.6%) | 52 (9.9%) | 4 (0.8%) | 524 (100.0%) |
| | 55–64 | 307 (42.5%) | 119 (16.5%) | 119 (16.5%) | 54 (7.5%) | 24 (3.3%) | 4 (0.6%) | 3 (0.4%) | 88 (12.2%) | 5 (0.7%) | 723 (100.0%) |
| | 65–74 | 246 (48.2%) | 96 (18.8%) | 53 (10.4%) | 45 (8.8%) | 11 (2.2%) | 4 (0.8%) | 2 (0.4%) | 46 (9.0%) | 7 (1.4%) | 510 (100.0%) |
| | 75+ | 241 (53.9%) | 89 (19.9%) | 25 (5.6%) | 14 (3.1%) | 26 (5.8%) | 7 (1.6%) | 6 (1.3%) | 35 (7.8%) | 4 (0.9%) | 447 (100.0%) |
| | **Total** | **992 (45.0%)** | **407 (18.5%)** | **314 (14.2%)** | **152 (6.9%)** | **67 (3.0%)** | **17 (0.8%)** | **14 (0.6%)** | **221 (10.0%)** | **20 (0.9%)** | **2204 (100.0%)** |
| Multiple Falls | 45–54 | 34 (47.2%) | 11 (15.3%) | 11 (15.3%) | 5 (6.9%) | 0 (0.0%) | 0 (0.0%) | 1 (1.4%) | 10 (13.9%) | 0 (0.0%) | 72 (100.0%) |
| | 55–64 | 60 (45.8%) | 25 (19.1%) | 21 (16.0%) | 5 (3.8%) | 5 (3.8%) | 0 (0.0%) | 0 (0.0%) | 12 (9.2%) | 3 (2.3%) | 131 (100.0%) |
| | 65–74 | 27 (39.1%) | 16 (23.2%) | 8 (11.6%) | 4 (5.8%) | 5 (7.2%) | 0 (0.0%) | 2 (2.9%) | 7 (10.1%) | 0 (0.0%) | 69 (100.0%) |
| | 75+ | 47 (62.7%) | 9 (12.0%) | 4 (5.3%) | 3 (4.0%) | 4 (5.3%) | 1 (1.3%) | 1 (1.3%) | 4 (5.3%) | 2 (2.7%) | 75 (100.0%) |
| | **Total** | **168 (48.4%)** | **61 (17.6%)** | **44 (12.7%)** | **17 (4.9%)** | **14 (4.0%)** | **1 (0.3%)** | **4 (1.2%)** | **33 (9.5%)** | **5 (1.4%)** | **347 (100.0%)** |

Multiple Falls Group: two or more falls reported in preceding year (No or Yes)

information [33], all of which can affect falls, have been reported. While we did not find rural-urban differences in fall rates across any age group in the CLSA baseline data, it is not known whether this finding will continue to be observed as data are examined over time.

Our findings indicate that several health conditions typically not thought of to be associated with falls, such as mood disorder, anxiety disorder, and hyperthyroidism are associated with higher likelihood of multiple falls. Chronic health conditions and diseases may directly, indirectly, and synergistically increase fall risk and fall rates such as through the direct effects of the condition or disease; indirectly with the health condition affecting other physiological systems resulting in decreased physical activity and/or causing muscle weakness, sensory or other impairments and causing poor balance; or a combination of direct and indirect effects. Previous research has suggested chronic diseases and multiple pathologies to be most important predictors of falls and more important than other risk factors such as polypharmacy [34], which was not explored in the current study.

While the results related to where falls occur based on age group and the higher likelihood of falls in individuals with some of the chronic conditions were not surprising, the low incidence of falls overall and lack of difference in fall rate between age groups were unexpected. Similar to Verma and colleagues' 2016 study of falls and fall-related injuries of adults in the United States of America [35], the number of fallers in the older age groups was lower than previously reported. Several studies have reported that about a third of community-dwelling adults 65 years of age and older report falling in one year. Much of the literature to date has focused on falls in the older adult population, while data on fall incidence among younger, working aged adults is rare [35]. Our findings and the work of Verma et al suggest falls

**Table 6. Frequency of different types of the most serious injury by age group and single/multiple falls (N = 419).**

| Number of falls | Age Group (years) | Most serious injury | | | | | | | | | | | |
|---|---|---|---|---|---|---|---|---|---|---|---|---|---|
| | | No serious injury | Sprain or strain | Bruises | Cuts | Discomfort | Hip fracture | Leg fracture | Arm or wrist fracture | Vertebral fracture | Head injury | Other | Total |
| Single fall | 45–54 | 5 (5.7%) | 32 (36.4%) | 8 (9.1%) | 8 (9.1%) | 7 (8.0%) | 1 (1.1%) | 4 (4.5%) | 11 (12.5%) | 2 (2.3%) | 7 (8.0%) | 3 (3.4%) | 88 (100.0%) |
| | 55–64 | 7 (5.7%) | 42 (34.1%) | 12 (9.8%) | 8 (6.5%) | 9 (7.3%) | 1 (0.8%) | 7 (5.7%) | 17 (13.8%) | 2 (1.6%) | 8 (6.5%) | 10 (8.1%) | 123 (100.0%) |
| | 65–74 | 7 (8.6%) | 31 (38.3%) | 8 (9.9%) | 4 (4.9%) | 5 (6.2%) | 2 (2.5%) | 6 (7.4%) | 7 (8.6%) | 1 (1.2%) | 2 (2.5%) | 8 (9.9%) | 81 (100.0%) |
| | 75+ | 5 (7.2%) | 15 (21.7%) | 12 (17.4%) | 3 (4.3%) | 8 (11.6%) | 3 (4.3%) | 3 (4.3%) | 10 (14.5%) | 0 (0.0%) | 4 (5.8%) | 6 (8.7%) | 69 (100.0%) |
| | **Total** | **24 (6.6%)** | **120 (33.2%)** | **40 (11.1%)** | **23 (6.4%)** | **29 (8.0%)** | **7 (1.9%)** | **20 (5.5%)** | **45 (12.5%)** | **5 (1.4%)** | **21 (5.8%)** | **27 (7.5%)** | **361 (100.0%)** |
| Multiple falls | 45–54 | 0 (0.0%) | 8 (53.3%) | 2 (13.3%) | 1 (6.7%) | 1 (6.7%) | 1 (6.7%) | | 0 (0.0%) | 0 (0.0%) | 2 (13.3%) | | 15 (100.0%) |
| | 55–64 | 1 (4.0%) | 10 (40.0%) | 4 (16.0%) | 3 (12.0%) | 1 (4.0%) | 1 (4.0%) | | 2 (8.0%) | 1 (4.0%) | 2 (8.0%) | | 25 (100.0%) |
| | 65–74 | 0 (0.0%) | 2 (22.2%) | 2 (22.2%) | 0 (0.0%) | 1 (11.1%) | 0 (0.0%) | | 0 (0.0%) | 0 (0.0%) | 4 (44.4%) | | 9 (100.0%) |
| | 75+ | 0 (0.0%) | 5 (55.6%) | 0 (0.0%) | 0 (0.0%) | 0 (0.0%) | 0 (0.0%) | | 2 (22.2%) | 1 (11.1%) | 1 (11.1%) | | 9 (100.0%) |
| | **Total** | **1 (1.7%)** | **25 (43.1%)** | **8 (13.8%)** | **4 (6.9%)** | **3 (5.2%)** | **2 (3.4%)** | | **4 (6.9%)** | **2 (3.4%)** | **9 (15.5%)** | | **58 (100.0%)** |

screening and identification of individuals at high risk of falls and fall-related injuries should be comprised of a more multi-faceted pragmatic approach across the adult lifespan; for example, considering falling and falls as a universal vital sign of potential functional or other health issues, regardless of age e.g., versus only people who are 60 or 65 years and older; condition or disease e.g., versus only individuals who have a neurological condition that affects balance or muscle strength; or health care context e.g., expanding beyond settings such as acute or long-term care to public health, ambulatory and mental health care settings.

Routine inquiring about falls and fall-related injuries, for younger aged adults or those without known risk factors or health conditions, is limited, as the vast majority of falls and falls screening evidence to date is focused on older adults. For example, the American Geriatrics Society and British Geriatrics Society published a clinical practice guideline on fall risk screening, assessment, and management, with recommendations to screen all adults aged 65 years and older for fall risk annually [36]; recently published world guidelines on falls prevention and management are focused on older adults [37]; and government initiatives also suggest people aged 65 years and older be asked about their fall risk annually [38, 39], or with a significant change in clinical status [39].

Incorporating questions about falls and fall-related injuries on an annual basis or with significant change in health status, as per the guidelines for older adults, into all clinical and health visits regardless of age of the individual and care context would serve as a basis to direct more robust screening and assessment as needed. Interventions and education to increase awareness implemented early on and earlier may be useful in preventing falls as the individual ages or the health condition progresses. While recognizing that fall prevention in older adults is a priority, targeting increased awareness about falls and fall risk identification and prevention across all age groups, as well as addressing and controlling chronic health conditions and diseases earlier may also be important and beneficial public health strategies.

**Table 7. Prevalence of chronic conditions and odds of single or multiple falls relative to participants who did not fall.**

| Chronic Condition | Count (%) | Missing Count (%) | Odds Ratio [95% CI] | |
|---|---|---|---|---|
| | | | **Single Fall** | **Multiple Falls** |
| Other long-term physical or mental condition | 21103 (41.1%) | 118 (0.2%) | **1.170 [1.074–1.274]** | **1.619 [1.313–1.996]** |
| High blood pressure or hypertension | 19203 (37.4%) | 207 (0.4%) | **1.107 [1.015–1.207]** | 1.113 [0.899–1.378] |
| Diabetes, borderline diabetes or blood sugar is high | 8863 (17.3%) | 129 (0.3%) | 1.090 [0.977–1.216] | **1.814 [1.435–2.292]** |
| Mood disorder | 8250 (16.1%) | 129 (0.3%) | **1.465 [1.319–1.627]** | **3.353 [2.702–4.163]** |
| Cancer | 7902 (15.4%) | 109 (0.2%) | 1.076 [0.960–1.206] | **1.332 [1.024–1.733]** |
| Osteoarthritis in the knee | 7927 (15.4%) | 458 (0.9%) | **1.476 [1.328–1.641]** | **2.393 [1.902–3.012]** |
| Osteoarthritis in one or both hands | 6840 (13.3%) | 386 (0.8%) | **1.428 [1.276–1.599]** | **2.837 [2.254–3.570]** |
| Migraine headaches | 6773 (13.2%) | 136 (0.3%) | **1.284 [1.142–1.443]** | **2.248 [1.764–2.865]** |
| Under-active thyroid gland | 6408 (12.5%) | 569 (1.1%) | **1.345 [1.196–1.511]** | **1.673 [1.280–2.187]** |
| Other type of arthritis | 6292 (12.3%) | 503 (1.0%) | **1.323 [1.175–1.491]** | **1.590 [1.207–2.096]** |
| Asthma | 6331 (12.3%) | 166 (0.3%) | **1.397 [1.243–1.571]** | **2.471 [1.938–3.151]** |
| Heart disease (including congestive heart failure, or CHF) | 5694 (11.1%) | 228 (0.4%) | 1.056 [0.926–1.205] | **1.962 [1.508–2.554]** |
| Bowel disorder | 4776 (9.3%) | 171 (0.3%) | **1.312 [1.147–1.501]** | **2.239 [1.705–2.941]** |
| Osteoporosis | 4698 (9.2%) | 341 (0.7%) | **1.639 [1.447–1.857]** | **2.820 [2.182–3.643]** |
| Osteoarthritis in the hip | 4586 (8.9%) | 416 (0.8%) | **1.487 [1.305–1.693]** | **2.966 [2.301–3.823]** |
| Urinary incontinence | 4388 (8.5%) | 125 (0.2%) | **1.547 [1.358–1.762]** | **2.461 [1.876–3.227]** |
| Anxiety disorder | 4157 (8.1%) | 146 (0.3%) | **1.424 [1.239–1.637]** | **3.345 [2.592–4.316]** |
| Intestinal or stomach ulcers | 3912 (7.6%) | 171 (0.3%) | **1.213 [1.044–1.409]** | **1.992 [1.470–2.699]** |
| Peripheral vascular disease or poor circulation in limbs | 3165 (6.2%) | 230 (0.4%) | **1.304 [1.112–1.528]** | **1.904 [1.362–2.663]** |
| Emphysema, chronic bronchitis, COPD, or chronic changes in lungs due to smoking | 3161 (6.2%) | 202 (0.4%) | **1.524 [1.309–1.773]** | **3.137 [2.361–4.170]** |
| Heart attack or myocardial infarction | 2778 (5.4%) | 179 (0.3%) | 0.854 [0.701–1.041] | **1.529 [1.043–2.240]** |
| Angina | 2473 (4.8%) | 190 (0.4%) | 0.937 [0.765–1.147] | **2.133 [1.501–3.032]** |
| Macular degeneration | 2155 (4.2%) | 257 (0.5%) | **1.225 [1.010–1.486]** | **1.794 [1.200–2.682]** |
| Rheumatoid arthritis | 2058 (4.0%) | 516 (1.0%) | 1.191 [0.972–1.458] | **2.273 [1.559–3.315]** |
| Experienced a ministroke or TIA | 1713 (3.3%) | 306 (0.6%) | **1.244 [1.003–1.542]** | **2.321 [1.551–3.472]** |
| Kidney disease or kidney failure | 1460 (2.8%) | 160 (0.3%) | 1.171 [0.921–1.488] | **2.760 [1.844–4.131]** |
| Over-active thyroid gland (hyperthyroidism) | 1190 (2.3%) | 568 (1.1%) | 0.778 [0.470–1.288] | **3.591 [1.927–6.693]** |
| Bowel incontinence | 1074 (2.1%) | 105 (0.2%) | **1.496 [1.165–1.922]** | **3.003 [1.922–4.689]** |
| Memory problem | 968 (1.9%) | 113 (0.2%) | **1.409 [1.069–1.859]** | **5.534 [3.822–8.013]** |

*(Continued)*

**Table 7.** (Continued)

| Chronic Condition | Count (%) | Missing Count (%) | Odds Ratio [95% CI] | |
|---|---|---|---|---|
| | | | **Single Fall** | **Multiple Falls** |
| Stroke or CVA | 912 (1.8%) | 144 (0.3%) | **1.567 [1.203– 2.040]** | **2.313 [1.349–3.966]** |
| Epilepsy | 488 (1.0%) | 98 (0.2%) | **1.804 [1.279– 2.545]** | **3.528 [1.920–6.482]** |
| Multiple sclerosis | 343 (0.7%) | 89 (0.2%) | **1.701 [1.118– 2.586]** | **5.969 [3.388– 10.514]** |
| Parkinsonism or Parkinson's Disease | 203 (0.4%) | 86 (0.2%) | 1.113 [0.588–2.107] | **3.544 [1.448–8.674]** |
| Dementia or Alzheimer's disease | 111 (0.2%) | 89 (0.2%) | 1.231 [0.539–2.812] | 2.596 [0.638–10.573] |

Values in bold indicate the odds ratio is statistically significantly different from 1 at the 95% confidence level.

Identifying individuals at high-risk of fall is complex, as evidenced by the ever-increasing number of falls screening and risk tools that have been developed for older adults specifically, people with various health conditions, as well as for specific care continuum contexts e.g., acute care, inpatient care, long-term care and rehabilitation facilities. Many individuals are referred to and engage in a fall prevention program (screening, assessment, intervention) after sustaining a fall-related injury, which may be too late to be consequential in the longer-term. Due to the significant morbidity, mortality, social and economic burden of falls and fall-related injuries in the Canadian population [7, 39–43], a primary versus secondary prevention approach to fall risk and falls across the lifespan is warranted.

## Limitations

This study was a cross-sectional study of CLSA baseline data and is thus representative of the CLSA participant population (i.e., versus representative of the full Canadian population). Cross-sectional data analysis does not provide a prospective analysis of causal relationships, and our analysis was limited by the type of data items collected. For example our data found younger participants were more likely to report falling on snow or ice, however we cannot delve further and analyze whether these falls occurred during sports activities versus when walking because of how the related question was asked. Our sample was comprised of largely urban-dwelling participants and was generally reflective of participants with higher socioeconomic status and level of education than the wider Canadian population and for whom CLSA study participation may have fewer barriers, introducing possible biases. The CLSA dataset does not include adult participants ages 20 to 44 years, thus our findings were limited to middle- and older- age groups.

Retrospective recording of falls is a low-cost, convenient, time-efficient, and widely used method [44]. However retrospective self-report of falls may not provide as accurate a picture as data collected prospectively or via direct measures; and may have introduced recall bias and led to under-reporting of falls and fall-related injuries. Further while seeking medical attention may be indicative of more serious fall, this may also reflect sociodemographic and personal factors that influence the likelihood of getting medical care after a fall [34].

While a strength of study is the use of a large, population-based dataset that aimed to be representative and utilized in-person interview to collect data, due to the large sample size statistical significance findings should be interpreted with some caution. Some community-dwelling population groups e.g., individuals living in the Canadian territories, and veterans are

excluded from the CLSA and representation of ethnicity and race is limited, decreasing the broader applicability of the findings.

## Conclusion

Approximately one in twenty adults in the CLSA baseline sample reported experiencing a fall in the preceding year, and about one in one hundred reported falling multiple times. Likelihood of falls was similar regardless of age or urban/rural residence, but age was associated with fall location or situation. Risk factors for single and multiple falls were consistent with previous literature, although some unexpected risk factors for multiple falls were identified (self-reported mood and anxiety disorders, hyperthyroidism). A more multifaceted approach, not based solely on older age, to screening, primary prevention, assessment and management of falls and fall-related risk factors, including chronic disease management across the lifespan is warranted to reduce the personal, social, and economic burden of falls and fall-related injuries.

## Acknowledgments

This research was made possible using the data collected by the Canadian Longitudinal Study on Aging (CLSA). This research has been conducted using the CLSA datasets Baseline Tracking Dataset version 3.4, Baseline Comprehensive Dataset version 4.0, under Application Number 171011. The CLSA is led by Drs. Parminder Raina, Christina Wolfson and Susan Kirkland.

We thank Shoshanna Green, Research Coordinator, University of Saskatchewan; and Katelyn Madigan, Research Assistant, McMaster University for their editorial assistance with the manuscript.

## Author Contributions

**Conceptualization:** Vanina P. M. Dal Bello-Haas, Megan E. O'Connell.

**Data curation:** Megan E. O'Connell.

**Formal analysis:** Megan E. O'Connell.

**Funding acquisition:** Vanina P. M. Dal Bello-Haas.

**Investigation:** Vanina P. M. Dal Bello-Haas, Megan E. O'Connell.

**Methodology:** Vanina P. M. Dal Bello-Haas, Megan E. O'Connell.

**Project administration:** Vanina P. M. Dal Bello-Haas.

**Supervision:** Vanina P. M. Dal Bello-Haas.

**Visualization:** Vanina P. M. Dal Bello-Haas, Megan E. O'Connell, Jake Ursenbach.

**Writing – original draft:** Vanina P. M. Dal Bello-Haas, Megan E. O'Connell, Jake Ursenbach.

**Writing – review & editing:** Vanina P. M. Dal Bello-Haas, Megan E. O'Connell, Jake Ursenbach.

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
