## [Decision Letter · Decision Letter 0]

5 Dec 2023

PONE-D-23-32442Decade-by-decade comparison of falls in adults living in urban and rural Canada: A cross-sectional analysis from the Canadian Longitudinal Study on AgingPLOS ONE

Dear Dr. Dal Bello-Haas,

Thank you for submitting your manuscript to PLOS ONE. After careful consideration, we feel that it has merit but does not fully meet PLOS ONE’s publication criteria as it currently stands. Therefore, we invite you to submit a revised version of the manuscript that addresses the points raised during the review process. Two reviewers made some comments on this paper. Please revise your manuscript concerning their comments.

We look forward to receiving your revised manuscript.

Kind regards,

Ryota Sakurai, Ph.D.

Academic Editor

PLOS ONE

Journal Requirements:

McMaster Institute of Research on Aging (MIRA) for Canadian Longitudinal Study on Aging (CLSA) data access.

This research was made possible using the data collected by the Canadian Longitudinal Study on Aging (CLSA). Funding for the Canadian Longitudinal Study on Aging (CLSA) is provided by the Government of Canada through the Canadian Institutes of Health Research (CIHR) under grant reference: LSA 94473 and the Canada Foundation for Innovation, as well as the following provinces, Newfoundland, Nova Scotia, Quebec, Ontario, Manitoba, Alberta, and British Columbia. This research has been conducted using the CLSA datasets Baseline Tracking Dataset version 3.4, Baseline Comprehensive Dataset version 4.0, under Application Number 171011. The CLSA is led by Drs. Parminder Raina, Christina Wolfson and Susan Kirkland.

We thank Shoshanna Green, Research Coordinator, University of Saskatchewan; and Katelyn Madigan, Research Assistant, McMaster University for their editorial assistance with the manuscript.

McMaster Institute of Research on Aging (MIRA) for Canadian Longitudinal Study on Aging (CLSA) data access.

5. Thank you for stating the following in the Competing Interests/Financial Disclosure* (delete as necessary) section: 

Co-author Megan O'Connell is a Member - Psychology Working Group, Canadian Longitudinal Study on Aging (CLSA) https://www.clsa-elcv.ca/

We note that one or more of the authors are employed by a commercial company: Psychology Working Group, Canadian Longitudinal Study on Aging. 

“The funder provided support in the form of salaries for authors, but did not have any additional role in the study design, data collection and analysis, decision to publish, or preparation of the manuscript. The specific roles of these authors are articulated in the ‘author contributions’ section.”

Reviewers' comments:

Reviewer's Responses to Questions

**Comments to the Author**

1. Is the manuscript technically sound, and do the data support the conclusions?

Reviewer #1: Yes

Reviewer #2: Yes

2. Has the statistical analysis been performed appropriately and rigorously? 

Reviewer #1: Yes

Reviewer #2: Yes

3. Have the authors made all data underlying the findings in their manuscript fully available?

Reviewer #1: Yes

Reviewer #2: Yes

4. Is the manuscript presented in an intelligible fashion and written in standard English?

Reviewer #1: Yes

Reviewer #2: Yes

5. Review Comments to the Author

Reviewer #1: Comment-(i)

Line 1

“Decade-by-decade comparison of falls in adults living in urban and rural Canada: A cross-sectional analysis from the Canadian Longitudinal Study on Aging”

See Comment-(viii). Also, the comparison between urban and rural areas is only one of the several axes of comparison, as well as the results on this axe were negative. My suggested title is then;

“Comparison across age groups of causes, circumstances, and consequences of falls among adults living in Canada: a cross-sectional analysis from the Canadian Longitudinal Study of Aging”

Comment-(ii)

Line18-20

Background:

Please briefly describe the purpose of the study.

Comment-(iii)

Comment deleted.

Comment-(iv)

Line24

“Falls were not related to rural residence or age, but those with memory problems or multiple sclerosis were more likely to fall. Higher likelihood of falls was also observed in people with chronic conditions not often thought to be associated with falls.”

Can these two sentences be combined into one? My suggestion is;

“Falls were not related to rural residence or age, but those with memory impairment, multiple sclerosis, as well as other chronic conditions such as xxx, yyy, zzz, not often thought to be associated with falls were also more likely to fall.”

Comment-(v)

Line 32

“A more universal approach to screening, primary prevention and management may reduce the personal, social, and economic burden of falls and fall-related injuries.”

What is the opposite concept of “universal approach” here? Please see also comment-(xxiv).

Comment-(vi)

Comment deleted.

Comment-(vii)

Comment deleted.

Comment-(viii)

Line 63

“The objectives of our research were to: 1) comprehensively examine fall profiles decade-by-decade e.g., 45 to 54 years, 55 to 64 years, 65 to 74 years, 75 to 85 years;…”

The term "decade-by-decade" is to me more reminiscent of changes by era, such as the 1980s, 1990s, 2000s... rather than by age group. How about expressions like “age-stratified analysis of…” or “comparison across age groups”?

Comment-(ix)

Table 1

I think it would be better to place related diseases closer together rather than in the order of frequency of diseases. For example, it is natural to place “Other type of arthritis” below other types. However, I do not strongly urge the author to do so, as he/she may have own ideas. Please consider it if a reviewer other than me has the same opinion as mine.

Comment-(x)

Line 142

“What is readily apparent and surprising is the low prevalence of falls.”

In the "Results" section, it is better to state only the facts in a straightforward manner. I propose to delete this sentence.

Comment-(xi)

Line 147

“Table 2 includes information about falls (no fall, single fall, multiple falls) with decade-by-decade comparisons.”

Again, perhaps a phrase such as "age group comparisons" would be more appropriate than "decade-by-decade comparisons"?

Comment-(xii)

Line 162

“The effect size estimate was small in magnitude for both the single fall and multiple falls groups, but only the larger single sized fall group had a significant chi-square.”

Please consider replacing this sentence with the following;

“The effect size estimate was small in magnitude for both the single fall and multiple falls groups, but only the single fall group had a significant chi-square.”

Comment-(xiii)

Line 163

“A weak association between where falls occurred and age was evident (Table 4), with older participants were more likely to fall inside the home than outside the home relative to younger participants.”

(xiii-i) “Weak” sounds subjective. “An association between...” would be more appropriate.

(xiii-ii) It seems necessary to state this more objectively by showing some statistical significance. For example, how about combining the two categories of “Inside Home” and “Outside home but inside a building”, making fall location binary data, and applying the Cochran-Armitage trend test?

Comment-(xiv)

Line 174

“Older participants who reported a single fall were more likely to report falling on snow or ice than were younger participants, but the converse was apparent for participants in the multiple falls group, with younger participants more likely to report falling on snow or ice.”

Question;

Do falls on snow or ice include those related to winter sports such as skiing or skating? I think this is an important point in interpreting the results.

Comment-(xv)

Line 168

“Age was moderately associated with how falls occurred for both the single fall group and the multiple fall group (Table 5).”

I suggest removing "moderately", since it seems to lack objectivity.

Comment-(xvi)

Line 177

“Older participants who were in the single fall group were more likely to report falling from furniture, however reported falls getting in and out of the shower and bath were infrequent.”

Instead, I suggest the following sentence;

“Older participants in the single fall group were more likely to report falling from furniture. Falls getting in and out of the shower and bath were infrequent regardless of age group.”

Comment-(xvii)

Line184

“The likelihood of having multiple falls versus a single fall was substantially higher (i.e., large magnitude of effect) for participants who reported being diagnosed with memory problems or with multiple sclerosis.”

The story is too complex to consider multiple falls in comparison to a single fall. I suppose the following sentence suffices;

“The likelihood of having multiple falls was substantially higher (i.e., large magnitude of effect) for participants who reported being diagnosed with memory problems or with multiple sclerosis.”

Comment-(xviii)

Line194

“These OR are reported in Table 7.”

Please provide a reference to Table 7 at the beginning of the paragraph, not at the end.

Comment-(xix)

Line 206

“Our findings indicate that several health conditions typically not thought of to be associated with falls had higher likelihood of multiple falls, such as mood disorder, anxiety disorder, and hyperthyroidism.”

Instead, I suggest the following sentence;

“Our findings indicate that several health conditions typically not thought of to be associated with falls, such as mood disorder, anxiety disorder, and hyperthyroidism are associated with higher likelihood of multiple falls.”

Comment-(xx)

Line 221

“Our findings and the work of Verma et al suggest falls screening and identification of individuals at high risk of falls and fall-related injuries should be comprised of a more universal and pragmatic approach across the adult lifespan; for example, falls as a vital sign of potential functional or other health issue, regardless of age, condition or disease, or health care context.”

About the last part, I understand that “age” can be disregarded, but wouldn't it be more in line with the context of this study to consider “condition” and “disease”?

Comment-(xxi)

Line 225

“Routine inquiring about falls and fall-related injuries, for younger aged adults or those without known risk factors or health conditions, is not being conducted.”

What country is this about? Do you mean that it is not done anywhere in the world?

Comment-(xxii)

Line 265

“Approximately one in twenty adults in the CLSA baseline sample reported experiencing a fall in the last year, and about one in one hundred reported falling multiple times.”

“Previous year” or “preceding year” rather than “last year” would be more appropriate for an objective description.

Comment-(xxiii)

Line 266

“Likelihood of falls occurring was similar regardless of age or urban/rural residence, but age was associated with fall location and activity.”

The last part sounds a bit strange to me. My suggestion is;

“Likelihood of falls was similar regardless of age or urban/rural residence, but age was associated with fall location or situation.”

Comment-(xxiv)

Line 269

“A more universal approach to screening, primary prevention, assessment and management of falls and fall-related risk factors, including chronic disease management across the lifespan is warranted to reduce the personal, social, and economic burden of falls and fall-related injuries.”

The meaning of “universal” sounds ambiguous.

Is it like “a more multifaceted approach that is not based solely on age” ?

Reviewer #2: Summary:

I would like to express my gratitude to the authors for their manuscript, which provides important insights into falls among middle-aged and older adult populations residing in both urban and rural regions of Canada. Its credibility is strongly supported by the utilization of a substantial dataset drawn from the Canadian Longitudinal Study on Aging (CLSA). The research presents some noteworthy findings that challenge prevailing preconceptions regarding fall-related factors, carrying substantial implications for the development of preventive measures and the allocation of healthcare resources. The study's concluding remarks underscore the imperative of adopting a comprehensive and universally applicable approach to the prevention and management of falls.

General comment:

Given the study exclusively includes 45+ years old adults (middle-aged and older adults), I recommend that authors clearly define or reconsider the use of the term 'adults' to avoid potential confusion throughout the manuscript. For example, information in ln. 42-43 indicate that adults would be defined as 20 years and older “Almost half of Canadians ages 20 years and older are living with a chronic condition”.

Specific comments:

Manuscript's title:

- The current title, 'The 'decade-to-decade'...', may benefit from a more explicit indication that the study pertains to age groups, specifically middle-aged and older adults.

Abstract:

- While the background information is relevant, it would be beneficial to explicitly state the study's key objectives in the abstract. This would provide readers with a clearer understanding of the study's focus, which extends beyond older individuals.

Introduction:

- I recommend addressing my earlier comment on defining the term 'adults' for clarity.

- Overall, the introduction effectively contextualizes the study and provides a clear foundation for the research objectives, helping readers understand the rationale and significance of the study.

Methods:

- In the brief outline, consider including the years when the data were collected or specifying the data collection period within the method section to provide clarity.

- To enhance reader understanding, it would be beneficial to clearly differentiate between 'age groups' and age as a continuous variable.

- In line 96, please expand 'M/F' to 'Male/Female'.

- For lines 99-102, adding a citation for the categorization and providing a brief in-text definition of rural areas would be helpful.

- Please specify the type of logistic regression used (e.g., binary/univariate logistic regression)

- In line 136, consider citing a reference.

- In line 138, I suggest including 'phi' to prepare readers for subsequent mentions.

- Ln. 129-130 “Logistic regression was used to explore whether demographic factors (age measured as a continuous variable), sex, education level, rural/urban) were predictors of fall status, that is no falls, a single fall, or multiple falls (reference group ‘no falls’).”

o Please, ensure consistency between the variables listed here and those reported in the results (e.g. results on sex and education level).

Results:

- Ensure that all tables can stand alone.

- Table 1. I suggest that you specify the time period for income, and add '$...CAD' for international readers.

- For Tables 3-5, clarify in a footnote how the variable 'multiple falls' was treated when falls occurred in different locations.

- Table 7. Please explain the bold font.

Discussion:

- The discussion effectively summarizes key findings, offers valuable insights and recommendations for addressing falls and fall-related injuries across the lifespan, and identifies potential areas for future research. I have couple of suggestions:

o In line 220, you mention 'younger, working-aged adults'. Consider finding a place, such as the limitations section, to clarify that this dataset does not include young adults (ages 20 to 44).

- Regarding reference no. 33, check if referring to a more recent paper, such as Montero-Odasso et al., 2022 (DOI: 10.1093/ageing/afac205), might provide more up-to-date information.

Conclusion:

- The study's conclusion closely matches its objectives. It successfully summarizes findings on falls, age [groups], urban/rural residence, and risk factors while highlighting the need for universal fall prevention across the lifespan.

Minor comments:

- Please address instances where periods are missing, and ensure that citations are positioned after punctuation marks.

- In line 112, there appears to be missing information in the sentence starting with 'Additional fall-related…'.

6. PLOS authors have the option to publish the peer review history of their article (what does this mean?). If published, this will include your full peer review and any attached files.

Reviewer #1: **Yes: **Narumi Kojima

Reviewer #2: No

---

## [Author Response · Author response to Decision Letter 0]

26 Jan 2024

I have uploaded a response to reviewers document, I do not understand what is required here that would be different. So I am cutting and pasting my letter in this space ....

January 16, 2024

Ryota Sakurai, Ph.D.

Academic Editor

PLOS ONE

Dear Dr. Sakurai,

On behalf of the co-authors, I would like to thank you and the Reviewers for their very thoughtful reviews, comments and feedback to enhance our paper. 

All comments received have been cut and paste and our responses to each comment, and found below. Our responses are highlighted in yellow in the revised manuscript document. 

Revisions within the manuscript have also been highlighted in yellow and are evident via track changes; and/or manuscript sections where the revisions were made are noted in this letter because page and line numbers have changed from the original submission, because of additions made, and page and line numbers differ between the marked-up and clean copies.

We believe the comments and feedback received and subsequent revisions have greatly improved our paper and we look forward to hearing from you soon.

Please note: We have revised the ACKNOWLEDGEMENTS AND FUNDING SECTIONS. We moved the following section to the FUNDING, similar to other CLSA publications within PLOSONE and in articles published as recently as 2023 [e.g., Khan D, Edgell H, Rotondi M, Tamim H. The association between shift work exposure and cognitive impairment among middle-aged and older adults: Results from the Canadian Longitudinal Study on Aging (CLSA). PLoS One. 2023 Aug 23;18(8):e0289718. doi: 10.1371/journal.pone.0289718]: 

Funding for the Canadian Longitudinal Study on Aging (CLSA) is provided by the Government of Canada through the Canadian Institutes of Health Research (CIHR) under grant reference: LSA 94473 and the Canada Foundation for Innovation, as well as the following provinces, Newfoundland, Nova Scotia, Quebec, Ontario, Manitoba, Alberta, and British Columbia.

Sincerely,

Vanina Dal Bello-Haas, PT, PhD

 

FEEDBACK RECEIVED

#1. Journal Requirements:

and 

We have reviewed the information found at the links and believe we have address the outstanding requirements as needed.

Per the data access and publication agreements we have signed with the Canadian Longitudinal Study on Aging we are not able to deposit the data.

Per the agreements we included the following statement within the Data Availability section, this remains the same, see page XX:

Data are available from the Canadian Longitudinal Study on Aging (www.clsa-elcv.ca) for researchers who meet the criteria for access to de-identified CLSA data.

McMaster Institute of Research on Aging (MIRA) for Canadian Longitudinal Study on Aging (CLSA) data access.

The funders had no role in the study. We have added the above statement above within the FUNDING section.

This research was made possible using the data collected by the Canadian Longitudinal Study on Aging (CLSA). Funding for the Canadian Longitudinal Study on Aging (CLSA) is provided by the Government of Canada through the Canadian Institutes of Health Research (CIHR) under grant reference: LSA 94473 and the Canada Foundation for Innovation, as well as the following provinces, Newfoundland, Nova Scotia, Quebec, Ontario, Manitoba, Alberta, and British Columbia. This research has been conducted using the CLSA datasets Baseline Tracking Dataset version 3.4, Baseline Comprehensive Dataset version 4.0, under Application Number 171011. The CLSA is led by Drs. Parminder Raina, Christina Wolfson and Susan Kirkland.

We thank Shoshanna Green, Research Coordinator, University of Saskatchewan; and Katelyn Madigan, Research Assistant, McMaster University for their editorial assistance with the manuscript.

We have revised the ACKNOWLEDGEMENTS AND FUNDING SECTIONS. We moved the following section to the FUNDING, similar to other CLSA publications within PLOSONE and in articles published as recently as 2023 [e.g., Khan D, Edgell H, Rotondi M, Tamim H. The association between shift work exposure and cognitive impairment among middle-aged and older adults: Results from the Canadian Longitudinal Study on Aging (CLSA). PLoS One. 2023 Aug 23;18(8):e0289718. doi: 10.1371/journal.pone.0289718]: 

Funding for the Canadian Longitudinal Study on Aging (CLSA) is provided by the Government of Canada through the Canadian Institutes of Health Research (CIHR) under grant reference: LSA 94473 and the Canada Foundation for Innovation, as well as the following provinces, Newfoundland, Nova Scotia, Quebec, Ontario, Manitoba, Alberta, and British Columbia. 

We have left the other statements within ACKNOWLEDGEMENTS.

See Page XX.

McMaster Institute of Research on Aging (MIRA) for Canadian Longitudinal Study on Aging (CLSA) data access.

There is no funding-related text in the manuscript.

We have added this information to the cover letter.

5. Thank you for stating the following in the Competing Interests/Financial Disclosure* (delete as necessary) section: 

Co-author Megan O'Connell is a Member - Psychology Working Group, Canadian Longitudinal Study on Aging (CLSA) https://www.clsa-elcv.ca/

We note that one or more of the authors are employed by a commercial company: Psychology Working Group, Canadian Longitudinal Study on Aging. 

Dr. Megan O’Connell is not employed by the Psychology Working Group, Canadian Longitudinal Study on Aging – her role is a non-paid service role. Dr. O’Connell has no commercial affiliation with the CLSA.

To be as encompassing as possible regarding any perceived Conflict of Interest, because Dr. O’Connell is a member of a Canadian Longitudinal Study on Aging Working Group, even though it is NOT a paid position and there is no commercial affiliation, we included her role and a [potential perceived] Conflict of Interest.

If the editors and journal staff deem this statement is not required, this statement can be removed.

See above. There has been NO financial support in the form of author’s salaries and/or research materials.

“The funder provided support in the form of salaries for authors, but did not have any additional role in the study design, data collection and analysis, decision to publish, or preparation of the manuscript. The specific roles of these authors are articulated in the ‘author contributions’ section.”

See above. This statement is not applicable. We have not included the statement.

See above. This statement is not applicable. We have not included the statement.

See above. This statement is not applicable. We have not included the statement.

See above. This statement is not applicable. We have not included the statement.

See above. This statement is not applicable. We have not included the statement.

We cannot provide repository information for the data. Per the data access and publication agreements we have signed with the Canadian Longitudinal Study on Aging we are not able to deposit the data.

Per the agreements we did add the following statement within the Data Availability section with the original submission, and this remains the same, see last page before the references:

Data are available from the Canadian Longitudinal Study on Aging (www.clsa-elcv.ca) for researchers who meet the criteria for access to de-identified CLSA data.

We do not have any Supporting Information files to include.

 

Reviewers' comments:

Reviewer's Responses to Questions

Comments to the Author

1. Is the manuscript technically sound, and do the data support the conclusions?

Reviewer #1: Yes

Reviewer #2: Yes

2. Has the statistical analysis been performed appropriately and rigorously?

Reviewer #1: Yes

Reviewer #2: Yes

3. Have the authors made all data underlying the findings in their manuscript fully available?

Reviewer #1: Yes

Reviewer #2: Yes

4. Is the manuscript presented in an intelligible fashion and written in standard English?

Reviewer #1: Yes

Reviewer #2: Yes

5. Review Comments to the Author

Reviewer #1: Comment-(i)

Line 1

“Decade-by-decade comparison of falls in adults living in urban and rural Canada: A cross-sectional analysis from the Canadian Longitudinal Study on Aging”

See Comment-(viii). Also, the comparison between urban and rural areas is only one of the several axes of comparison, as well as the results on this axe were negative. My suggested title is then;

“Comparison across age groups of causes, circumstances, and consequences of falls among adults living in Canada: a cross-sectional analysis from the Canadian Longitudinal Study of Aging”

We have made this change; and we have also made additional revisions to the title per Reviewer #2’s comments, see below.

Comment-(ii)

Line18-20

Background:

Please briefly describe the purpose of the study.

We have added this information to the Abstract background section.

Comment-(iii)

Comment deleted.

Comment-(iv)

Line24

“Falls were not related to rural residence or age, but those with memory problems or multiple sclerosis were more likely to fall. Higher likelihood of falls was also observed in people with chronic conditions not often thought to be associated with falls.”

Can these two sentences be combined into one? My suggestion is;

“Falls were not related to rural residence or age, but those with memory impairment, multiple sclerosis, as well as other chronic conditions such as xxx, yyy, zzz, not often thought to be associated with falls were also more likely to fall.”

We have made this change.

Comment-(v)

Line 32

“A more universal approach to screening, primary prevention and management may reduce the personal, social, and economic burden of falls and fall-related injuries.”

What is the opposite concept of “universal approach” here? Please see also comment-(xxiv).

We have revised the statement in the Abstract to clarify and better reflect our intended thoughts and perspectives.

Comment-(vi)

Comment deleted.

Comment-(vii)

Comment deleted.

Comment-(viii)

Line 63

“The objectives of our research were to: 1) comprehensively examine fall profiles decade-by-decade e.g., 45 to 54 years, 55 to 64 years, 65 to 74 years, 75 to 85 years;…”

The term "decade-by-decade" is to me more reminiscent of changes by era, such as the 1980s, 1990s, 2000s... rather than by age group. How about expressions like “age-stratified analysis of…” or “comparison across age groups”?

We have changed to “age groups …” [in the Abstract and throughout the manuscript].

Comment-(ix)

Table 1

I think it would be better to place related diseases closer together rather than in the order of frequency of diseases. For example, it is natural to place “Other type of arthritis” below other types. However, I do not strongly urge the author to do so, as he/she may have own ideas. Please consider it if a reviewer other than me has the same opinion as mine.

We have revised Table 1.

Comment-(x)

Line 142

“What is readily apparent and surprising is the low prevalence of falls.”

In the "Results" section, it is better to state only the facts in a straightforward manner. I propose to delete this sentence.

The sentence has been deleted, see Results section.

Comment-(xi)

Line 147

“Table 2 includes information about falls (no fall, single fall, multiple falls) with decade-by-decade comparisons.”

Again, perhaps a phrase such as "age group comparisons" would be more appropriate than "decade-by-decade comparisons"?

We have replaced with “age group”, see Table 2.

Comment-(xii)

Line 162

“The effect size estimate was small in magnitude for both the single fall and multiple falls groups, but only the larger single sized fall group had a significant chi-square.”

Please consider replacing this sentence with the following;

“The effect size estimate was small in magnitude for both the single fall and multiple falls groups, but only the single fall group had a significant chi-square.”

We have made this change, see Results section.

Comment-(xiii)

Line 163

“A weak association between where falls occurred and age was evident (Table 4), with older participants were more likely to fall inside the home than outside the home relative to younger participants.”

(xiii-i) “Weak” sounds subjective. “An association between...” would be more appropriate.

We have made this change, removing the word weak, see Results section.

(xiii-ii) It seems necessary to state this more objectively by showing some statistical significance. For example, how about combining the two categories of “Inside Home” and “Outside home but inside a building”, making fall location binary data, and applying the Cochran-Armitage trend test?

Thank you for this interesting suggestion. We kept the categories separate. Although both are considered “indoors”, one setting is a familiar setting (“Inside Home”) and the other setting is considered an unfamiliar setting (Outside the home but inside a building”). An unfamiliar environment may be an important extrinsic falls risk factor, worthy of exploring. Keeping the variables separate is also in keeping with other research using large data sets e.g., Moreland BL, Kakara R, Haddad YK, Shakya I, Bergen G. A Descriptive Analysis of Location of Older Adult Falls That Resulted in Emergency Department Visits in the United States, 2015. Am J Lifestyle Med. 2020 Aug 7;15(6):590-597. doi: 10.1177/1559827620942187.

Comment-(xiv)

Line 174

“Older participants who reported a single fall were more likely to report falling on snow or ice than were younger participants, but the converse was apparent for participants in the multiple falls group, with younger participants more likely to report falling on snow or ice.”

Question;

Do falls on snow or ice include those related to winter sports such as skiing or skating? I think this is an important point in interpreting the results.

We agree this would be an interesting and important consideration. Unfortunately, the CLSA data does not allow us to delve into this. Below is how the question is asked of participants and entered into the CLSA database.

We have added this as an example to the Limitations section.

FAL_Q05

FAL_HOW_TRM

How did your fall happen? READ LIST, CODE ONLY ONE RESPONSE

Fell while standing or walking .............................. 01

Fell on stairs or steps ........................................... 02

Fell while exercising (except walking) .................. 03

Fell from height of greater than 1 meter or

3 feet (for example, ladder, tree, roof) ............... 04

[ONLY ASK IF FAL_Q04/FAL_WHERE_TRM=1 OR 2]

Fell from furniture (for example, bed, chair) ......... 05

[ONLY ASK IF FAL_Q04/FAL_WHERE_TRM=1 OR 2]

Fell while getting in or out of the bathtub ............. 06

[ONLY ASK IF FAL_Q04/FAL_WHERE_TRM=1 OR 2]

Fell while getting in or out of the shower ............. 07

[ONLY ASK IF FAL_Q04/FAL_WHERE_TRM=3]

Fell on snow or ice ............................................... 08

FAL_HOW_OTSP_TRM Other (please specify: ______________) ........... 97

[DO NOT READ] Don’t know/No answer ............ 98

[DO NOT READ] Refused ................................... 99

Comment-(xv)

Line 168

“Age was moderately associated with how falls occurred for both the single fall group and the multiple fall group (Table 5).”

I suggest removing "moderately", since it seems to lack objectivity.

We have made this change, see Results section.

Comment-(xvi)

Line 177

“Older participants who were in the single fall group were more likely to report falling from furniture, however reported falls getting in and out of the shower and bath were infrequent.”

Instead, I suggest the following sentence;

“Older participants in the single fall group were more likely to report falling from furniture. Falls getting in and out of the shower and bath were infrequent regardless of age group.”

We have made this change, see Results section.

Comment-(xvii)

Line184

“The likelihood of having multiple falls versus a single fall was substantially higher (i.e., large magnitude of effect) for participants who reported being diagnosed with memory problems or with multiple sclerosis.”

The story is too complex to consider multiple falls in comparison to a single fall. I suppose the following sentence suffices;

“The likelihood of having multiple falls was substantially higher (i.e., large magnitude of effect) for participants who reported being diagnosed with memory problems or with multiple sclerosis.”

We have made this change, see Results section.

Comment-(xviii)

Line194

“These OR are reported in Table 7.”

Please provide a reference to Table 7 at the beginning of the paragraph, not at the end.

We have made this change, see Results section.

Comment-(xix)

Line 206

“Our findings indicate that several health conditions typically not thought of to be associated with falls had higher likelihood of multiple falls, such as mood disorder, anxiety disorder, and hyperthyroidism.”

Instead, I suggest the following sentence;

“Our findings indicate that several health conditions typically not thought of to be associated with falls, such as mood disorder, anxiety disorder, and hyperthyroidism are associated with higher likelihood of multiple falls.”

We have made this change, see Discussion section.

Comment-(xx)

Line 221

“Our findings and the work of Verma et al suggest falls screening and identification of individuals at high risk of falls and fall-related injuries should be comprised of a more universal and pragmatic approach across the adult lifespan; for example, falls as a vital sign of potential functional or other health issue, regardless of age, condition or disease, or health care context.”

About the last part, I understand that “age” can be disregarded, but wouldn't it be more in line with the context of this study to consider “condition” and “disease”?

We have revised the statement to clarify and better reflect our intended thoughts and perspectives, see Discussion section.

Comment-(xxi)

Line 225

“Routine inquiring about falls and fall-related injuries, for younger aged adults or those without known risk factors or health conditions, is not being conducted.”

What country is this about? Do you mean that it is not done anywhere in the world?

We were wanting to convey that a lifespan approach to falls screening, falls interventions/programs seems to be limited – literature and web-based searches confirm this. The focus of research and initiatives has been and seems to continue to be on older adults and this seems applicable globally. For example, one can only find documents that state “older adults over 65 years should be considered a high risk for falls” versus documents that state “everyone you see clinically should be asked about falls”.

We have revised the statement to clarify and better reflect our intended thoughts and perspectives, see Discussion section.

Comment-(xxii)

Line 265

“Approximately one in twenty adults in the CLSA baseline sample reported experiencing a fall in the last year, and about one in one hundred reported falling multiple times.”

“Previous year” or “preceding year” rather than “last year” would be more appropriate for an objective description.

We have changed “last year” to “preceding year”, see Conclusion section.

Comment-(xxiii)

Line 266

“Likelihood of falls occurring was similar regardless of age or urban/rural residence, but age was associated with fall location and activity.”

The last part sounds a bit strange to me. My suggestion is;

“Likelihood of falls was similar regardless of age or urban/rural residence, but age was associated with fall location or situation.”

We have made this change, see Conclusion section.

Comment-(xxiv)

Line 269

“A more universal approach to screening, primary prevention, assessment and management of falls and fall-related risk factors, including chronic disease management across the lifespan is warranted to reduce the personal, social, and economic burden of falls and fall-related injuries.”

The meaning of “universal” sounds ambiguous.

Is it like “a more multifaceted approach that is not based solely on age” ?

 We have made this change, and made additional revisions to better convey our thoughts and perspectives, see Conclusion section.

Reviewer #2: Summary:

I would like to express my gratitude to the authors for their manuscript, which provides important insights into falls among middle-aged and older adult populations residing in both urban and rural regions of Canada. Its credibility is strongly supported by the utilization of a substantial dataset drawn from the Canadian Longitudinal Study on Aging (CLSA). The research presents some noteworthy findings that challenge prevailing preconceptions regarding fall-related factors, carrying substantial implications for the development of preventive measures and the allocation of healthcare resources. The study's concluding remarks underscore the imperative of adopting a comprehensive and universally applicable approach to the prevention and management of falls.

Thank you.

General comment:

Given the study exclusively includes 45+ years old adults (middle-aged and older adults), I recommend that authors clearly define or reconsider the use of the term 'adults' to avoid potential confusion throughout the manuscript. For example, information in ln. 42-43 indicate that adults would be defined as 20 years and older “Almost half of Canadians ages 20 years and older are living with a chronic condition”.

We have addressed this comment by changing from adults to people, individuals, Canadians etc, as applicable, throughout the manuscript.

Specific comments:

Manuscript's title:

- The current title, 'The 'decade-to-decade'...', may benefit from a more explicit indication that the study pertains to age groups, specifically middle-aged and older adults.

See response to Reviewer 1 comment above as well. We have revised the Title to address both Reviewers’ comments.

Abstract:

- While the background information is relevant, it would be beneficial to explicitly state the study's key objectives in the abstract. This would provide readers with a clearer understanding of the study's focus, which extends beyond older individuals.

We have added this information, see Abstract.

Introduction:

- I recommend addressing my earlier comment on defining the term 'adults' for clarity.

As above - We have addressed this comment by changing from adults to people, individuals, Canadians, etc, as applicable, in the Abstract and throughout the manuscript.

- Overall, the introduction effectively contextualizes the study and provides a clear foundation for the research objectives, helping readers understand the rationale and significance of the study.

Thank you.

Methods:

- In the brief outline, consider including the years when the data were collected or specifying the data collection period within the method section to provide clarity.

We have added this information, see Methods section.

- To enhance reader understanding, it would be beneficial to clearly differentiate between 'age groups' and age as a continuous variable.

- In line 96, please expand 'M/F' to 'Male/Female'.

We have made this change, see Methods section.

- For lines 99-102, adding a citation for the categorization and providing a brief in-text definition of rural areas would be helpful.

We have added this information and added citations, see Methods section.

- Please specify the type of logistic regression used (e.g., binary/univariate logistic regression)

We have added this information, see Methods section.

- In line 136, consider citing a reference.

We have added a reference, see Methods section.

- In line 138, I suggest including 'phi' to prepare readers for subsequent mentions.

We have added this information and citation, see Methods section.

- Ln. 129-130 “Logistic regression was used to explore whether demographic factors (age measured as a continuous variable), sex, education level, rural/urban) were predictors of fall status, that is no falls, a single fall, or multiple falls (reference group ‘no falls’).”

o Please, ensure consistency between the variables listed here and those reported in the results (e.g. results on sex and education level).

We have made revisions and believe we have captured all instances of previous non-alignment, see Methods section.

Results:

- Ensure that all tables can stand alone.

- Table 1. I suggest that you specify the time period for income, and add '$...CAD' for international readers.

Table 1 has been revised, see Results section.

- For Tables 3-5, clarify in a footnote how the variable 'multiple falls' was treated when falls occurred in different locations.

We have added a footnote to Tables 3 to 5.

- Table 7. Please explain the bold font.

We have added an explanation to Table 7.

Discussion:

- The discussion effectively summarizes key findings, offers valuable insights and recommendations for addressing falls and fall-related injuries across the lifespan, and identifies potential areas for future research. 

Thank you.

I have couple of suggestions:

o In line 220, you mention 'younger, working-aged adults'. Consider finding a place, such as the limitations section, to clarify that this dataset does not include young adults (ages 20 to 44).

We have made this addition to the Limitations section.

- Regarding reference no. 33, check if referring to a more recent paper, such as Montero-Odasso et al., 2022 (DOI: 10.1093/ageing/afac205), might provide more up-to-date information.

Conclusion:

- The study's conclusion closely matches its objectives. It successfully summarizes findings on falls, age [groups], urban/rural residence, and risk factors while highlighting the need for universal fall prevention across the lifespan.

Thank you.

Minor comments:

- Please address instances where periods are missing, and ensure that citations are positioned after punctuation marks.

We have reviewed the manuscript and made the necessary changes and believe we have captured all applicable instances.

- In line 112, there appears to be missing information in the sentence starting with 'Additional fall-related…'.

This has been corrected, see Fall data sub-section (Methods).

6. PLOS authors have the option to publish the peer review history of their article (what does this mean?). If published, this will include your full peer review and any attached files.

Do you want your identity to be public for this peer review? For information about this choice, including consent withdrawal, please see our Privacy Policy.

Reviewer #1: Yes: Narumi Kojima

Reviewer #2: No

---

## [Decision Letter · Decision Letter 1]

21 Feb 2024

Comparison across age groups of causes, circumstances, and consequences of falls among individuals living in Canada: A cross-sectional analysis of participants aged 45 to 85 years from the Canadian Longitudinal Study on Aging

PONE-D-23-32442R1

Dear Dr. Dal Bello-Haas,

We’re pleased to inform you that your manuscript has been judged scientifically suitable for publication and will be formally accepted for publication once it meets all outstanding technical requirements.

Kind regards,

Ryota Sakurai, Ph.D.

Academic Editor

PLOS ONE

Additional Editor Comments (optional):

I re-invited only one reviewer who made major comment and he satisfied your revision. I also believe that the manuscript improved with comments from the reviewers. Thanks for providing important information to the PlosOne.

Reviewers' comments:

Reviewer's Responses to Questions

**Comments to the Author**

1. If the authors have adequately addressed your comments raised in a previous round of review and you feel that this manuscript is now acceptable for publication, you may indicate that here to bypass the “Comments to the Author” section, enter your conflict of interest statement in the “Confidential to Editor” section, and submit your "Accept" recommendation.

Reviewer #1: All comments have been addressed

2. Is the manuscript technically sound, and do the data support the conclusions?

Reviewer #1: Yes

3. Has the statistical analysis been performed appropriately and rigorously? 

Reviewer #1: Yes

4. Have the authors made all data underlying the findings in their manuscript fully available?

Reviewer #1: Yes

5. Is the manuscript presented in an intelligible fashion and written in standard English?

Reviewer #1: Yes

6. Review Comments to the Author

Reviewer #1: As I wrote in the previous round, the Abstract does not describe the purpose of the study.

Please summarize at the last part of Background in one sentence like "The study aims to investigate ....", what you wrote in lines 62-66.

7. PLOS authors have the option to publish the peer review history of their article (what does this mean?). If published, this will include your full peer review and any attached files.

Reviewer #1: **Yes: **Narumi Kojima

---

## [Editor Report · Acceptance letter]

5 Mar 2024

PONE-D-23-32442R1 

PLOS ONE

Dear Dr. Dal Bello-Haas, 

I'm pleased to inform you that your manuscript has been deemed suitable for publication in PLOS ONE. Congratulations! Your manuscript is now being handed over to our production team.

Kind regards, 

on behalf of

Dr. Ryota Sakurai 

Academic Editor

PLOS ONE